# Misfolding of Lysosomal α-Galactosidase a in a Fly Model and Its Alleviation by the Pharmacological Chaperone Migalastat

**DOI:** 10.3390/ijms21197397

**Published:** 2020-10-07

**Authors:** Hila Braunstein, Maria Papazian, Gali Maor, Jan Lukas, Arndt Rolfs, Mia Horowitz

**Affiliations:** 1The Shmunis School of Biomedicine and Cancer Research, Life Sciences, Tel Aviv University, Ramat Aviv 69978, Israel; brown_hila@walla.co.il (H.B.); mari.papazian@gmail.com (M.P.); galifit@gmail.com (G.M.); 2Department of Neurology, Translational Neurodegeneration Section “Albrecht-Kossel“, University Medical Center Rostock, Rostock 18051, Germany; jan.lukas@med.uni-rostock.de; 3Center for Transdisciplinary Neurosciences Rostock (CTNR), University Medical Center Rostock, University of Rostock, 18051 Rostock, Germany; 4Centogene AG, 18055 Rostock, Germany; arndt.rolfs@med.uni-rostock.de

**Keywords:** Fabry disease 1, misfolding 2, UPR 3, ERAD 4, migalastat 5

## Abstract

Fabry disease, an X-linked recessive lysosomal disease, results from mutations in the *GLA* gene encoding lysosomal α-galactosidase A (α-Gal A). Due to these mutations, there is accumulation of globotriaosylceramide (GL-3) in plasma and in a wide range of cells throughout the body. Like other lysosomal enzymes, α-Gal A is synthesized on endoplasmic reticulum (ER) bound polyribosomes, and upon entry into the ER it undergoes glycosylation and folding. It was previously suggested that α-Gal A variants are recognized as misfolded in the ER and undergo ER-associated degradation (ERAD). In the present study, we used *Drosophila melanogaster* to model misfolding of α-Gal A mutants. We did so by creating transgenic flies expressing mutant α-Gal A variants and assessing development of ER stress, activation of the ER stress response and their relief with a known α-Gal A chaperone, migalastat. Our results showed that the A156V and the A285D α-Gal A mutants underwent ER retention, which led to activation of unfolded protein response (UPR) and ERAD. UPR could be alleviated by migalastat. When expressed in the fly’s dopaminergic cells, misfolding of α-Gal A and UPR activation led to death of these cells and to a shorter life span, which could be improved, in a mutation-dependent manner, by migalastat.

## 1. Introduction

In misfolding diseases such as lysosomal storage diseases (LSDs) there is chronic retention of misfolded proteins in the endoplasmic reticulum (ER), which leads to their ER-associated degradation (ERAD) and causes ER stress and activation of the ER stress response in cells, known as the unfolded protein response (UPR) [1,2].

Our lab showed in the past that in Gaucher disease (GD), resulting from accumulation of glucosylceramides due to mutations in the *GBA1* gene, encoding acid-β-glucocerebrosidase (GCase), the mutant variants are retained in the ER and activate the UPR [3,4,5,6,7,8]. Using *Drosophila melanogaster* as an animal model, Maor et al. showed that expression of mutant human GCase in the dopaminergic cells of the fly activated UPR, which led to death of these cells, to motoric disabilities and to shorter life span. The phenotype could be rescued by treatment with the known pharmacological chaperone ambroxol [3,8].

In the present study we used *Drosophila melanogaster* as an in-vivo model to analyze misfolding of α-Gal A, linked with Fabry disease, and its associated UPR, and to test whether we can improve UPR-associated pathology by applying a pharmacological chaperone.

Fabry disease is the second most common lysosomal disorder, inherited as an X-linked recessive disease. It results from mutations in the *GLA* gene, encoding alpha galactosidase A (α-Gal A, EC 3.2.1.22, NM_000169.2), leading to accumulation of globotriaosylceramide (GL-3) and its acylated form lyso-GL3 (lyso-Gb3) [9,10,11,12,13] in various types of cells, including vascular endothelial cells, podocytes, cardiomyocytes, arterial smooth muscle cells and kidney cells, peripheral and central nervous systems, skin and eyes [10,11,12,13]. Death occurs mainly because of renal failure along with premature myocardial infarction and strokes [11,14,15].

Fabry disease is broadly divided into “classic” and “late onset” phenotypes. Patients with the classic phenotype of Fabry disease are usually males with significantly low or undetectable (<3% of mean normal) α-Gal A activity, who present multiple organ manifestations. Patients with the late-onset phenotype have varied ages of onset and clinical manifestations, with typical cardiac and renal symptoms [16,17]. Fabry disease females are heterozygous and present varying degrees of symptoms, ranging from asymptomatic to severe [18,19,20]. This phenotypic is due, most probably, to X-inactivation. There are more than 1000 mutations in the *GLA* gene known to be associated with Fabry disease [17,21]. It was estimated that 35–50% of patients with Fabry disease have mutations that are amenable to migalastat therapy [17,22]. The amenability of *GLA* mutations to migalastat therapy is determined by an in-vitro assay, in which human embryonic kidney (HEK) 293 cells are transfected with individual *GLA*-containing DNA plasmids, with and without migalastat. Migalastat-amenable mutations are defined as those that present absolute increases of ≥3% over wild-type α-Gal A activity in the presence of 10 μM migalastat [17,22].

Enzyme replacement therapy (ERT) based on intravenous administration of recombinant human enzyme has been available for Fabry disease since 2001 (Raplagal, Shire Human Genetic Therapies AB; Fabrazyme, Sanofi Genzyme) [23,24,25,26]. One main shortcoming of ERT is limited tissue penetrance. Central nervous system manifestations such as cerebrovascular complications and neuropathic pain in Fabry disease cannot be addressed due to the blood–brain barrier, which is not penetrated by ERT. Another shortcoming is the risk of an immune response with potentially neutralizing antibodies generated against the therapeutic enzymes [16,24,27]. Treatment with a pharmacological chaperone is another option for Fabry disease patients. Pharmacological chaperones are small molecules with the ability to cross the blood–brain barrier, which bind misfolded proteins in the ER to allow their correct folding and trafficking via the secretory pathway to the lysosomes. In the case of a lysosomal enzyme, the pharmacological chaperone dissociates in the lysosome from the enzyme, which binds to its substrate and catalyzes its hydrolysis according to its residual activity [9,28,29]. One approved chaperone for the treatment of Fabry disease patients is migalastat (AT1001, DGJ-1-deoxygalactonojirimycin, commercial name Galafold) [17]. Migalastat is an analogue of α-Gal A substrate that binds to the active site of the enzyme [30,31]. Treatment with migalastat was shown to increase enzyme activity in cell culture and in mice expressing amenable mutations [32,33,34,35,36,37].

In the present study, we established transgenic flies harboring mutant (A156V, A285D) α-Gal A variants to model their misfolding and to address the effect of a known pharmacological chaperone on this misfolding. Mutant or WT *GLA* mutant or WT *GLA* cDNAs, coupled to a yeast upstream activating sequence (UAS), which is inactive in the fly, were introduced into normal flies Expression of the transgenes was controlled by the UAS/GAL4 system. In this system, the GAL4 gene is placed under the control of a native gene promoter. When expressed, GAL4 binds and activates the UAS, which is coupled to the gene of interest. Thus, expression of a target gene (a *GLA* variant in the present study) is achieved by the presence of active GAL4 [38].

Our results strongly indicated that the mutant variants, but not the WT α-Gal A variant, were retained in the ER and underwent ERAD. Their retention in the ER activated the UPR machinery. When expression of the mutant α-Gal A variants was driven in the dopaminergic cells of the fly, misfolding and UPR activation led to death of these cells and to premature death of the flies, which could be improved, in a mutation dependent manner, by migalastat.

## 2. Results

### 2.1. Alpha-Gal a Is ER Retained and Undergoes ERAD

Three fly lines harboring different variants of the *GLA* gene were generated: a WT human α-Gal A, a mutated human A156V α-Gal A variant and a mutated human A285D variant. The two latter are considered classical mutations, the A156V variant has 4.3% in vitro residual activity while the A285D mutant has no detectable activity [21]. All were introduced as cDNAs, coupled to a yeast upstream activating sequence (UAS), which is inactive in the fly, and binds the transcription factor GAL4. Thus, UAS-linked transgenes can be expressed in specific cell types under the control of a *Drosophila* promoter linked to the GAL4 sequence [39].

Western blot analysis of lysates prepared from flies expressing the different α-Gal A variants under the ubiquitous daughterless-GAL4 (DaGAL4) driver [40] depicted lower levels of α-Gal A in flies expressing the mutant forms of α-Gal A (A156V and A285D) in comparison to flies expressing the WT variant (Figure 1A,B). Moreover, the level of the more severe A285D mutant protein was lower than that of the A156V variant (Figure 1A). The appearance of a major band in the WT sample, with a minor upper band, two bands in the A156V lysate and only the upper one in the A285D mutant, suggested that they represent different glycosylation states and cellular localization. Thus, we assumed that the upper band was ER retained protein, while the lower band represented the mature lysosomal α-Gal A form. To confirm our assumption, we treated the lysates with endoglycosidase-H (endo-H). Endo-H is a specific endo-glycosidase that cleaves a high mannose (more than four residues) *N*-glycan complex and not a mature counterpart [41,42]. Since high mannose structures (with 8–9 mannose residues in the N-glycan trees) are mostly found on glycoproteins present in the ER, this fraction of glycoproteins is endo-H sensitive. However, once glycoproteins arrive to the cis-Golgi network, five mannose residues are removed from their *N*-glycans, rendering them with only four to three mannoses on the *N*-glycan tree [43,44]. The oligosaccharide chain with four to three mannoses is not recognized by endoH. Hence, this fraction of glycoproteins, which is mostly lysosomal, is endo-H resistant. All three α-Gal A proteins showed sensitivity to endo-H cleavage, indicating that in all three variants, there was an ER retained endo-H sensitive protein. The endo-H sensitive fraction was increased with mutation severity. While ~17% of the WT protein was endo-H sensitive, ~46% and ~96% were endo-H sensitive in the A156V and the A285D mutants, respectively (Figure 1C,D). The results indicated that most of the A285D protein was ER retained, while a significant fraction of the A156V mutant variant was able to exit the ER. A small fraction of WT α-Gal A resided in the ER. Thus, the results of the endo-H resistance experiment confirmed that the upper α-Gal A band (seen in Figure 1A) represented the ER retained, most probably the unfolded form of the enzyme, while the lower protein was a mature lysosomal α-Gal A.

To test whether mutant α-Gal A variants undergo ERAD due to their misfolding, and due to sensitivity of flies to treatment with proteasome inhibitors like bortezomib or MG132, we performed the experiment in tissue culture. HEK293T cells, transfected with plasmids expressing the three different α-Gal A variants, were treated with MG132 after which their lysates were subjected to Western blot analysis. MG132 is a synthetic, membrane permeable, peptidyl aldehyde that effectively blocks the proteolytic activity of the S28 proteasome [45,46,47]. Therefore, treatment with MG132 leads to an increase in the amount of proteins that are otherwise degraded by ERAD. The results (Figure 1E,F) showed that the amount of mutant A156V and of the A285D α-Gal A proteins increased by MG132 treatment. There were 2.2- and 2.1-fold increases in the amount of A156V and A285D variants, respectively (Figure 1E,F), while the increase in the amount of the WT α-Gal A variant was non-significant. The results strongly indicate that mutant α-Gal A variants underwent ERAD (Appendix A).

In order to examine the effect of chaperone treatment on the expression of the α-Gal A variants in transgenic flies, α-Gal A-expressing flies were treated with different concentrations of migalastat for 22 days post-eclosion. Western blots of the proteins indicated that there was an increase in stability of WT α-Gal A in flies treated with 50 μM migalastat (Figure 2A,B). There was a 2.5- and a 3-fold increase in the total amount of the A156V α-Gal A following 10 μM and 50 μM chaperone treatment, respectively, and elevation in the lysosomal fraction of the mutant protein, indicating that migalastat is able to bind the protein and to assist in its folding and trafficking from the ER to the lysosomes (Figure 2C,D). Moreover, even in the A285D-expressing flies, there was a 2.5-fold increase in the total amount of α-Gal A, as well as appearance of a small lysosomal fraction, following 20 μM migalastat treatment (Figure 2E,F).

### 2.2. UPR Activation in Transgenic α-Gal a Expressing Flies

Our results strongly indicated that when expressed in flies, mutant α-Gal A variants were misfolded, were retained in the ER and underwent ERAD.

ER retention of misfolded proteins leads to ER stress and to activation of UPR. To examine UPR activation in flies expressing different mutant α-Gal A variants, RNA was extracted from 22 day old flies and was subjected to qRT-PCR. Changes in the levels of known UPR-induced fly genes were monitored [8]. The results showed activation of the tested UPR markers: Hsc-70-3 (the fly BiP homolog), spliced Xbp1 and ATF4 (Figure 3A). In order to test the effect of migalastat on UPR, flies, collected at the day of eclosion, were treated with different concentrations of the chaperone for 22 days, after which RNA was extracted and subjected to real-time quantitative polymerase chain reaction (qRT-PCR) analysis. The results (Figure 3B–E) strongly indicated that only the A156V α-Gal A mutation was responsive to the chaperone treatment. There was a decrease in the level of the tested UPR parameters: Hsc-70-3, spliced Xbp1 and ATF4 in flies expressing the A156V α-Gal A mutant after 22 days of migalastat treatment (Figure 3D). The highest concentration used, 50 μM, was proven to be the most effective. There was no significant change in UPR parameters in flies expressing the A285D α-Gal A variant (Figure 3E). There was a decrease in the level of spliced Xbp1 in flies expressing the WT α-Gal A only at 50 μM (Figure 3C) with no accompanied change in other genes. We, therefore, believe it does not reflect a real decrease in UPR. Interestingly, though there was a small increase in the lysosomal fraction of the A285D mutant (Figure 2E,F), it was not significant enough to cause a decrease in UPR.

### 2.3. Climbing Ability of Transgenic α-Gal a Expressing Flies

Negative geotaxis (climbing) of the flies, commonly used for assaying motor deficits as a result of misexpression of neuron-specific proteins [48], was employed to assess the neural dysfunction caused to the flies by expression of mutant α-Gal A in their dopaminergic cells. We monitored the climbing ability of the flies at ages of 15- and 22-days post-eclosion. The results presented significant locomotion dysfunction in flies expressing both A156V and A285D α-Gal A mutants compared to flies expressing the WT *GLA* variant or control flies (Figure 4A,B).

Climbing ability of flies, treated with different concentrations of migalastat, was evaluated as well. The results (Figure 4A,B) showed a significant improvement in climbing ability of the A156V mutant α-Gal A-expressing flies following chaperone treatment in all the tested concentrations. There was a slight change in the climbing ability of flies expressing the A285D mutation at day 15 post-eclosion; however, it did not reach statistical significance and was not recapitulated at day 22 post-eclosion.

### 2.4. Death of Dopaminergic Cells in Brains of GLA Flies

Thus far we have shown that mutant α-Gal A variants underwent ERAD, and that their ER retention led to UPR activation. This UPR activation, which affects the motor skills of the flies, could be significantly alleviated in flies expressing the A156V α-Gal A mutant variant by migalastat treatment.

To further explore UPR activation in the flies, we followed the death of their dopaminergic cells. To do that, we quantified the amount of tyrosine hydroxylase (TH), as a marker of dopaminergic cells, in heads of aging flies. Fluorescence intensity of TH was analyzed in the posterior region of the brain, where ~70 dopaminergic cells occupy very distinct areas [49]. The results showed significantly decreased fluorescence intensity in brains of 22 day old mutant flies. We did not observe a significant difference in flies expressing WT α-Gal A following chaperone treatment at 10–20 µM concentrations, with a 40% increase in the presence of 50 µM migalastat. This increase is in line with the increased amount of WT α-Gal A in the presence of 50 µM migalastat (Figure 2A,B). Fluorescence intensity was significantly increased in brains of the A156V α-Gal A-expressing flies. In contrast, the increase in fluorescence intensity was mild and only statistically significant at the 50 µM concentration in flies expressing the A285D mutant α-Gal A (Figure 5A,B).

To verify our results, we analyzed TH levels in head lysates of non-treated (Figure 6 A,B) and treated (Figure 6C–H) flies by Western blotting. The results strongly indicated that while 22 days treatment with migalastat had no effect in flies expressing the WT α-Gal A variant (Figure 6C,D), it had a significant effect on the amount of TH produced in the brains of A156V α-Gal A-expressing flies, which is a measure of surviving dopaminergic cells (Figure 6E,F). There was also a significant (though lower than that seen in the A156V α-Gal A-expressing flies) increase in the amount of TH in flies expressing the A285D mutant α-Gal A variant (Figure 6G,H).

### 2.5. Survival of Transgenic α-Gal a Expressing Flies

Life span of flies expressing the different α-Gal A variants was followed, with and without migalastat treatment. The results showed premature death of transgenic flies expressing mutant α-Gal A variants, in comparison to flies expressing the WT human α-Gal A (Figure 7A). Furthermore, the flies expressing the A285D α-Gal A mutation presented earlier death in comparison to flies expressing the A156V α-Gal A mutation (Figure 7A). In order to examine the effect of chaperone treatment on life span, α-Gal A-expressing flies were treated with different concentrations of migalastat from eclosion to death. The use of migalastat at different concentrations did not alter life span of flies expressing the human WT α-Gal A protein (Figure 7B) but significantly prolonged the life span of flies expressing the A156V α-Gal A mutant (Figure 7C). There was a small change in the survival of flies expressing the A285D α-Gal A mutation treated with 50 μM migalastat, which should not be ignored (Figure 6D). Interestingly 50% survival was also significantly increased for both A156V and A285D mutations (Figure 6E).

## 3. Discussion

Mutations in the *GLA* gene cause Fabry disease, an X-linked lysosomal disease characterized by progressive accumulation of GL-3 in cells. The accumulation leads to tissue damage and eventual organ failure. There are more than 1000 known mutations in the *GLA* gene [36,50]. Most of the mutations disrupt the hydrophobic core of the protein, presumably leading to protein misfolding and degradation in the ER [31,51,52,53]. Thus, Fabry disease is primarily a protein misfolding lysosomal disease.

In the present study we used *Drosophila melanogaster* to model misfolding of Fabry disease-associated mutant α-Gal A variants. We did so by creating transgenic flies expressing WT and mutant α-Gal A variants and assessing development of ER stress and activation of the ER stress response, and their relief with a known α-Gal A chaperone, migalastat. We tested two classical human mutant *GLA* variants, A156V, known to have 4.3% in vitro enzyme activity of WT, and the A285D α-Gal A mutation, with no residual enzymatic activity [21,52,54]. Our results clearly showed that the A156V and the A285D mutant proteins were misfolded and ER-retained. Their ER retention led to UPR activation, which could be reduced by migalastat treatment. Migalastat, also known as AT1001, is an active site-specific chaperone. It was first found as a potent competitive inhibitor of α-Gal A. However, it effectively enhances α-Gal A activity in Fabry-derived cells, when given at concentrations lower than that usually required for intracellular inhibition of the enzyme [31]. Migalastat is the first pharmacological chaperone approved for treating Fabry disease patients with known responsive mutations [17].

We also documented ERAD of the tested mutant α-Gal A variants. Thus, the level of two mutant proteins, A156V and the A285D, were elevated in transfected HEK293T cells, in the presence of the proteasome inhibitor MG132. By decreasing UPR parameters, we could achieve an increase in the number of TH containing dopaminergic cells in the brain, an improvement in motor abilities and expanded life span of the mutant flies, which were more significant in the A156V containing flies.

Our results corroborated published studies on the responsiveness of the different Fabry mutations to migalastat, arguing that they are misfolded. In an early report, Ishii et al. showed that in COS-1 cells transfected with plasmids expressing the R301Q or Q279E, there was a fraction of α-Gal A that was aggregated at the top of the gel and had no enzyme activity. The authors argued that the aggregate originated from the ER [55]. Using confocal microscopy, Yam et al. showed that following chaperone treatment, ER retained mutant α-Gal A successfully trafficked to the lysosomes and degraded GL-3 in fibroblasts that derived from Fabry disease patients. The authors suggested that migalastat reduced BiP binding to α-Gal A in the ER and allowed its trafficking [56,57]. Ishii et al. documented protein stabilization by Western blot analysis of several α-Gal A mutations, including the A156V variant, following either treatment with the ER mannosidase I inhibitor, kifunensine [58] or with the proteasomal inhibitor, lactacystin [59], in COS-7 cells transfected with plasmids expressing the tested mutations. The results indicated protein misfolding and ERAD of mutant α-Gal A variants. In addition, increase in activity following migalastat treatment was shown [60].

A large number of publications documented the effect of migalastat on stabilization and increase in activity of α-Gal A [32,61,62]. Improvement in enzyme activity of different mutant α-Gal A was shown in HEK293-transfected cells treated with either migalastat [21,63,64] or migalastat and ambroxol [65], the known pharmacological chaperone of glucocerebrosidase, deficient in Gaucher disease [66]. There was a synergistic effect when treating cells with both chaperones [65]. The described results strongly indicated ERAD of mutant α-Gal A variants.

Currently, there is a mouse model in which the endogenous gene was knocked out, and it expresses a human mutant R301Q α-Gal A cDNA under the human α-Gal A promoter [67]. While ERAD and UPR were not specifically shown in this model, decreased accumulation of GL-3 was documented following its treatment with migalastat [67,68]. These results strongly indicated misfolding of the R301Q mutant α-Gal A variant in the ER and its folding and trafficking to the lysosomes in the presence of the chaperone.

It is of note that De Francesco et al. ruled out apoptotic death associated with ER stress and UPR by evaluating caspase 4 and UPR genes in peripheral blood mononuclear cells derived from Fabry patients [69].

To summarize, in the present work we were able to document, using a fly model expressing α-Gal A variants, that the A156V and the A285D mutant variants are misfolded, are retained in the ER, undergo ERAD and activate the UPR. UPR activation by the A156V amenable mutation can be relieved by the pharmacological chaperone migalastat. On the other hand, treatment of the A285D-expressing flies with migalastat does not alleviate UPR parameters.

## 4. Materials and Methods

### 4.1. Antibodies

The antibodies used in the present project are as follows. Primary antibodies: rabbit polyclonal anti-tyrosine hydroxylase antibodies AB152 (Millipore, MA, USA); mouse monoclonal anti-Myc antibody (Cell Signaling Technology, Beverly, MA, USA); mouse monoclonal anti-actin antibody (Sigma-Aldrich, Rehovot, Israel); rabbit polyclonal anti-ERK antibodies (Santa Cruz Biotechnology, CA, USA). Secondary antibodies: horseradish peroxidase-conjugated goat anti-mouse antibodies; horseradish peroxidase-conjugated goat anti-rabbit antibodies, all from Jackson ImmunoResearch Laboratories, West Grove, PA, USA).

### 4.2. Fly Strains

Control strain in all experiments was Oregon-R obtained from the Bloomington *Drosophila* Stock Center, Indiana University, Bloomington, IN, USA). Transgenic flies, harboring pUAST-MycHis-WT *GLA* on the second chromosome, pUAST-MycHis-A156V *GLA* and pUAST-MycHis-A285D *GLA* on the third chromosome, were established by BestGene Inc. (Chino Hills, CA, USA). Da-GAL4 and Ddc-GAL4 were from Bloomington Stock Center (Indiana University, Bloomington, IN, USA). Strains were maintained on standard cornmeal-molasses medium at 25 °C.

### 4.3. Cell Lines and Transfections

HEK293T cells (ATCC^®^ CRL-11268™) were grown in Dulbecco’s modified Eagle’s medium (DMEM DMEM; Gibco, purchased from Biological industries, Beit-Haemek, Israel), supplemented with 10% FCS (Beit-Haemek, Israel) at 37 °C in the presence of 5% CO2. Cells were transfected using calcium phosphate solutions. A mixture of DNA in 250 μL of 250 mM CaCl2 was dropped into a tube containing HBSX2 solution (50 mM Hepes, 280 mM NaCl, 1.5 mM Na2HPO4, pH 7.09) and incubated for 20 min at RT. The mixture was then added dropwise to sub-confluent cells. Forty-eight hours later, cell lysates were prepared for SDS–PAGE and Western blot analysis.

### 4.4. MG132 (Carbobenzoxy-l-leucyl-l-leucyl-l-leucinal) Treatment

HEK293T-transfected cells were treated with 25 µM of MG132 24 h post transfection (Calbiochem, San Diego, CA, USA) for 20 h.

### 4.5. Plasmid Preparation

Gibson assembly (New England Biolabs, Beverly, MA, USA) was employed to create the pUAST-MycHis or the pcDNA4-MycHis-B plasmids, containing the different *GLA* cDNAs. The inserts were amplified from pcDNA3.1/V5-His-*GLA* (WT/A156V/A285D) plasmids (described in [21]), with primers shown in Table 1. PCR was executed in 20 µL containing 30 ng of plasmid DNA, 4 µl of 5× ISO buffer, 0.4 mM dNTPS, 3% DMSO, 20 units/mL of fusion polymerase (New England, Bio Labs, Beverly, MA, USA) and 10 pM each of forward and reverse primers. Thermal cycling conditions were 30 s at 98 °C, then thirty-five cycles of 98 °C (10 s), 68 °C (1 min) and 72 °C (1 min), following by 10 min at 72 °C for final extension. PCR reactions were carried out in Eorff Mastercycler EP Gradient S (Eorf, Hamburg, Germany). PCR products were separated on 1% agarose gels, and inserts were purified using “RBC Bioscience” kits, according to the manufacturer’s instructions. The PCR products contained XhoI sequences on both sides (See Table 1 for primer sequences). The vector plasmids pUAST-MycHis or pcDNA4-MycHis-B were linearized with the restriction enzyme XhoI. Assembly and transformation were performed using a Gibson Assembly Cloning Kit (New England, Bio Labs, Beverly, MA, USA), following the manufacturer’s instructions.

### 4.6. RNA Preparation

For RNA extraction from flies, adult flies were frozen in liquid nitrogen and then homogenized in TRIzol^®^ Reagent (Life Technologies, Carlsbad, CA, USA), according to the manufacturer’s instructions.

### 4.7. cDNA Preparation and qRT-PCR

Two micrograms of RNA were reverse transcribed with M-MLV reverse transcriptase (Promega Corporation, CA, USA), using oligo dT primer in a total volume of 20 μL at 42 °C for 60 min. Reactions were stopped by incubation at 70 °C for 15 min. Three microliters of cDNA were used for qRT-PCR. qRT-PCR was performed using power SYBR green QPCR mix reagent kit (Applied Biosystems, Foster City, CA, USA) Rotor-Gene 6000. The reaction mixture contained 50% QPCR mix, 300 nM of forward primer and 300 nM of reverse primer, in a final volume of 10 μL. Thermal cycling conditions were 10 min at 95 °C, 40 cycles of: 95 °C (10 s) 60 °C (20 s) and 72 °C (20 s). Relative gene expression was determined by Ct value. The list of primers used for qRT-PCR is shown in Table 2.

### 4.8. Endonuclease-H Sensitivity

Endonuclease-H (endo-H) sensitivity was tested essentially as described elsewhere [7]. Briefly, cell lysates, containing 80 µg of total protein, were subjected to an overnight incubation with endo-H (New England Biolabs, Beverly, MA, USA), according to the manufacturer’s instructions.

### 4.9. SDS–PAGE and Western Blotting

For each preparation, either confluent cells or 10 or more flies were homogenized in NP-40 lysis buffer (20 mM Tris HCL pH 7.5, 100 mM NaCl, 1 mM MgCl_2_, 5 Mm EDTA and 0.5% NP-40) containing protease inhibitors (10 μg/mL leupeptin, 10 μg/mL aprotinin and 0.1 mM PMSF, all from Sigma-Aldrich, St. Louis, MO, USA). Samples containing the same amount of protein were electrophoresed through 10% SDS–PAGE and electroblotted onto a nitrocellulose membrane (Schleicher and Schuell BioScience, Keene, NH, USA), which was interacted with the appropriate antibodies. The blots were developed and analyzed by ChemiDoc™ XRS (Bio-Rad laboratories, GmbH, Munich, Germany).

### 4.10. Immunofluorescence and Confocal Microscopy

Brains of adult flies were fixed with 4% paraformaldehyde for 60 min. Following rinsing with PBT (1× PBS supplemented with 0.3% Triton X-100), primary antibody, diluted in BBT (1× PBS supplemented with 0.1% BSA, 0.1% Tween-20 and 250 mM NaCl) was added for overnight incubation at 4 °C with shaking. Following three washes with PBT, secondary antibodies were added and were incubated under shaking for 2 h at room temperature. After three washes with PBT, the brain samples were mounted with Galvanol mounting reagent (Mowiol 4-88, Calbiochem, CA, USA). Slides were visualized using an LSM510 Meta (ZEISS) confocal microscope. For quantitative studies, Z-projections of confocal sections (exposed and processed identically) were analyzed. Images of a given experiment were exposed and processed identically, unless otherwise detailed. Captured images were analyzed using ImageJ software. The ImageJ software ((NIH, Bethesda, MD, USA) automatically subtracts background staining. Pixel intensity (in arbitrary units) was used to quantify fluorescence in the indicated experiments. Data was statistically evaluated using one-way ANOVA with post-hoc Dunnett test.

### 4.11. Chaperone Treatment

Eighty microliters of 1-deoxygalactonojirimycin hydrochloride (DGJ/AT1001/migalastat) purchased from Sigma-Aldrich, Israel, at concentrations of 10 µM, 20 µM or 50 μM were poured on top of 10 mL food-containing vials, which were kept at room temperature for at least 1 day before use. Chaperone was diluted in DDW from a 10 mM stock solution.

### 4.12. Climbing Assay

Climbing behavior of adult flies was measured using a countercurrent apparatus essentially as described elsewhere [70]. Briefly, groups of 30 flies (both males and females) were given 10 min to adapt in the starting tube, which could slide along the apparatus, and then 20 s to move upwards against gravity to the upper frame’s tube. The top frame of tubes was then shifted to the right so that the start tube came into register with a second bottom tube, and flies, which successfully climbed up, were tapped down again, falling into tube 2. The upper frame was then returned to the left, and the flies were once again allowed to climb into the upper tube. After five runs, the number of the flies in each tube was counted. For each time point, at least four cohorts from each genotype were scored. The Climbing index (CI) was calculated using the following formula: CI (the weighted mean) =  Σ(*mn_m_*)/N; *m*–number of test vial, *nm*–number of flies in the *m^th^* vial, N–total number of flies. CI ranged from 1 (min) to 6 (max).

### 4.13. Survival Assay

For each fly strain, 10 vials, each containing 5 males and 5 females (total of 100 flies), was maintained on food from day one post-eclosion. Flies were flipped into tubes containing fresh food every second day, and deaths were recorded. Kaplan–Meier was used to plot survival using the XLSTAT software. Results are presented relative to the initial number of flies in each vial.

### 4.14. Statistical analysis

Parametric statistical tests were used for all comparisons. Data, obtained from 3–6 independent experiments (in each experiment, 10–30 flies were used), were expressed as mean ± standard error of the mean (SEM). Comparisons between the groups were performed using one-way, two-way or three-way ANOVA according to the number of variants, followed by either post-hoc Dunnett test or t-test. The tests used for the different experiments are detailed in each figure legend. All statistical analyses were preformed using the SPSS software (IBM Corp., Armonk, NY, USA). Kaplan–Meier analysis was performed using the XLSTAT software (Addinsoft Inc., New York, NY, USA).

## Figures and Tables

**Figure 1 ijms-21-07397-f001:**
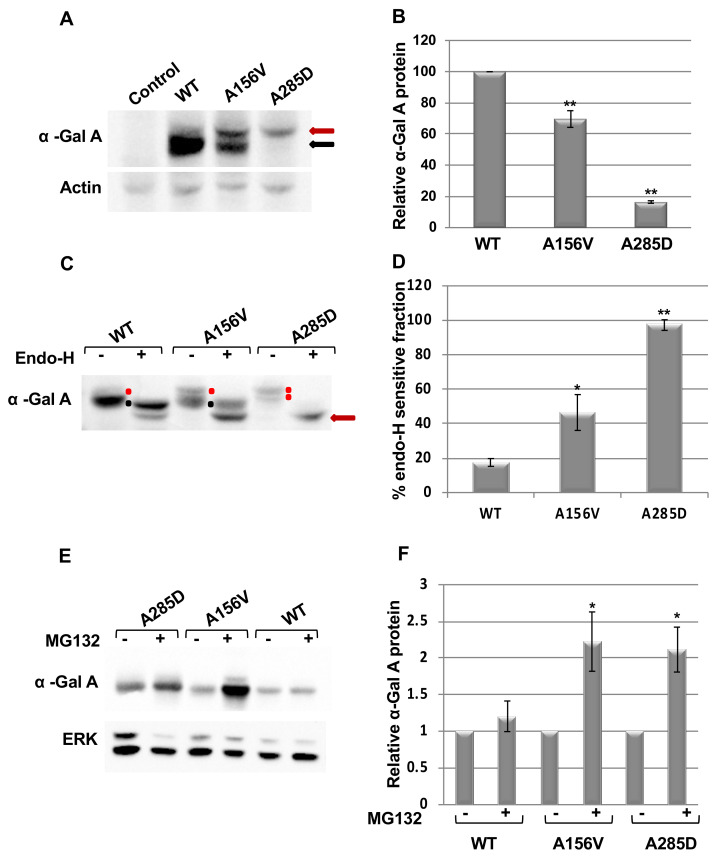
Alpha-Gal A expression in *Drosophila melanogaster*. (**A**) Lysates prepared from 15 day old flies were subjected to sodium dodecyl sulphate polyacrylamise gel electrophoresis (SDS–PAGE) and interaction with anti-Myc antibody, to identify α-Gal A and anti actin antibodies, as a loading control. The red arrow indicates the higher molecular weight, endoplasmic reticulum (ER) retained fraction of the protein; the black arrow indicates the lower molecular weight lysosomal fraction of the protein. (**B**) Quantification of results as seen in A of three independent experiments (mean ± standard error of mean (SEM)). To quantify the results, α-Gal A intensity in each lane was divided by that of actin in the same lane, and the number obtained for wild type (WT) was considered to be 100. Statistical analysis included one-way ANOVA followed by one-sample t-test. (**C**) Lysates prepared as in A were treated with endo-H and analyzed by Western blotting. The red dot indicates the higher molecular weight, ER retained fraction of the protein before endo-H treatment; the red arrow indicates the ER retained fraction of the protein after endo-H treatment; and the black dot marks the lower molecular weight, endo-H resistant lysosomal fraction of the protein. (**D**) Quantification of results as shown in (**C**). The amount of endo-H sensitive α-Gal A fraction in each treated lane was divided by the total amount of protein in the same lane. The results are the mean ± SEM of five independent experiments. Statistical analysis included one-way ANOVA followed by post-hoc Dunnett test. (**E**) Twenty-four hours after transfection of HEK293T cells with pcDNA4-MycHis-α-Gal A plasmids (human WT, A156V, A285D), they were treated with 25 μM MG132 for 20 h, and their lysates were subjected to Western blotting and interaction with anti-Myc antibody, to identify α-Gal A and anti ERK antibodies, as a loading control. (**F**) Quantification of results as shown in (**E**). To quantify the results, α-Gal A intensity in each lane was divided by that of ERK in the same lane, and the number obtained for every variant without treatment was considered to be 1. The results are the mean ± SEM of three independent experiments. Statistical analysis included one-sample t-test. Significance: * *p* < 0.05; ** *p* < 0.01. The expression of the α-Gal A mutant variants was under the daughterless GAL4 driver (DaGAL4).

**Figure 2 ijms-21-07397-f002:**
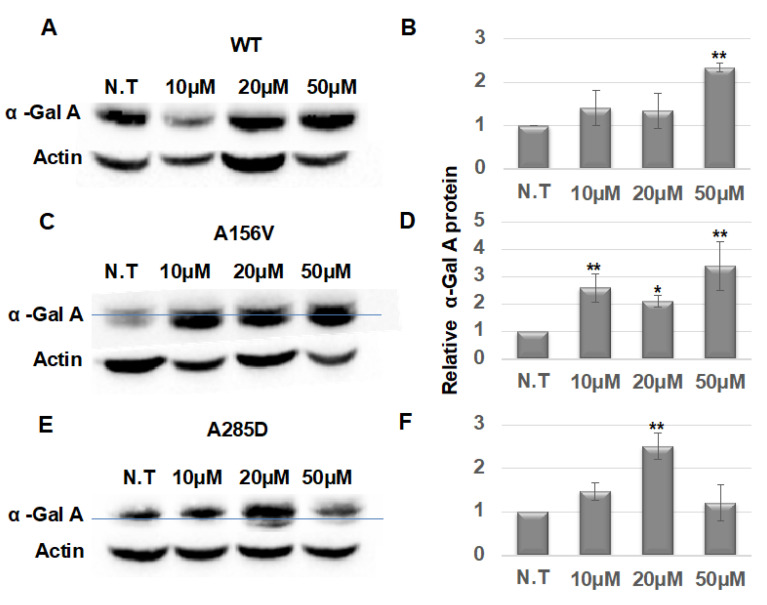
Alpha-Gal A expression following migalastat treatment. (**A**,**C**,**E**) Lysates prepared from 22 day old flies, carrying the different α-Gal A variants, as indicated in the graph, treated with different migalastat concentrations, were subjected to SDS–PAGE and interaction with anti-Myc antibody, to identify α-Gal A and anti actin antibodies, as a loading control. Blue lines separate the higher molecular weight ER retained fraction of the protein from the lower molecular weight lysosomal fraction (when applicable). (**B**,**D**,**F**) Quantification of results as shown in (**A**,**C**,**E**), respectively. To quantify the results, α-Gal A intensity in each lane was divided by that of actin in the same lane, and the number obtained for N.T. in each experiment was considered to be 1. The results are the mean ± SEM of three independent experiments. The expression of the α-Gal A mutant variants was under the daughterless GAL4 driver (DaGAL4). Statistical analysis included two-way ANOVA (to determine significant differences between genotypes’ reactions to concentrations) followed by one-way ANOVA with post-hoc Dunnett t-tests. Significance: * *p* < 0.05; ** *p* < 0.01. N.T: Non-Treated.

**Figure 3 ijms-21-07397-f003:**
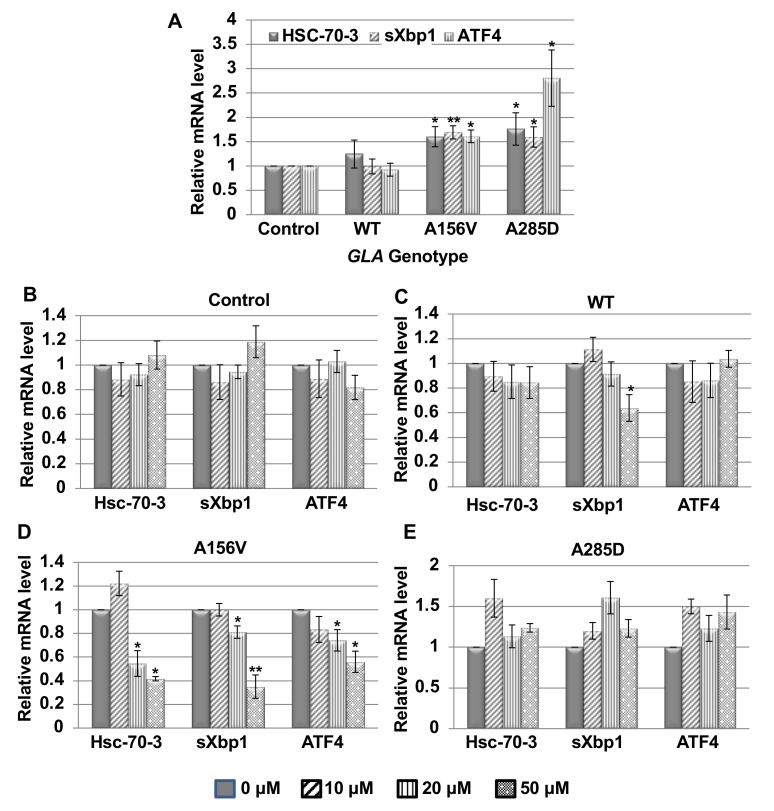
Unfolded protein response (UPR) activation in *GLA* transgenic flies and its alleviation following migalastat treatment. RNA was isolated from untreated (**A**) or from migalastat treated flies (**B**–**E**), expressing the different α-Gal A variants, as indicated in the graph. The extracted RNA was subjected to real-time quantitative polymerase chain reaction (qRT-PCR) with primers specific for Hsc-70-3, spliced Xbp1 (sXbp1) and ATF4 mRNAs (as specified in Table 2), representing the three UPR arms. The expression of the α-Gal A mutant variants was under the daughterless GAL4 driver (DaGAL4). The results are the mean ± SEM of five independent experiments. Each bar represents the results of five different experiments. Statistical analysis included two-way ANOVA (to show the significance of the reaction of the different genotypes to different migalastat concentrations), followed by one-sample t-test. Significance: * *p* < 0.05; ** *p* < 0.01.

**Figure 4 ijms-21-07397-f004:**
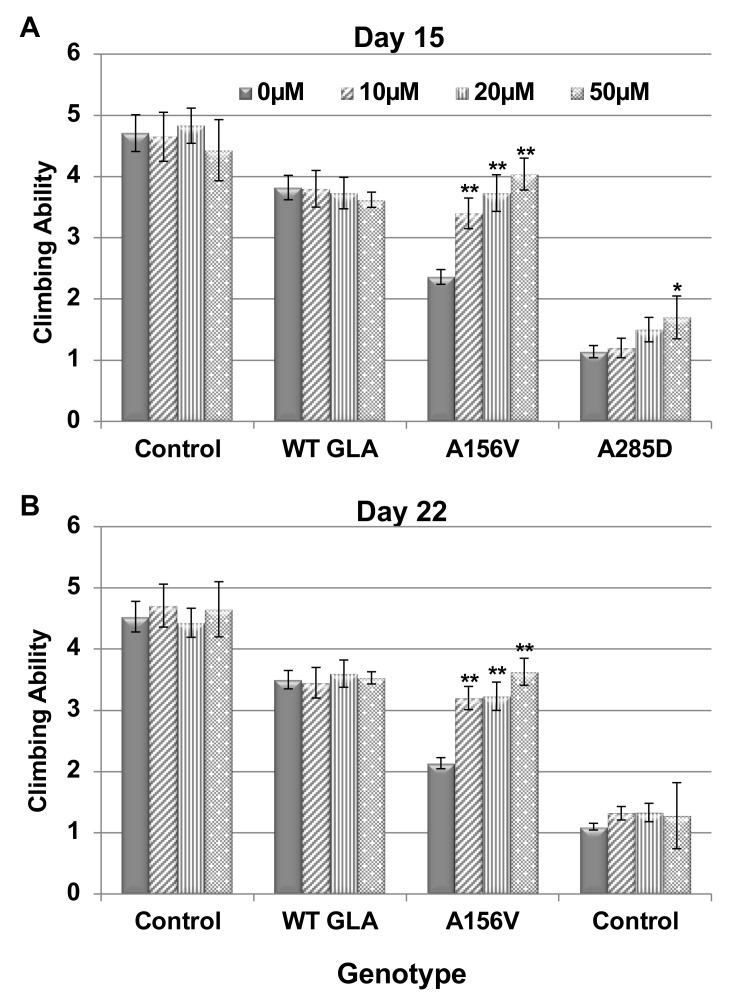
Climbing ability of *GLA* transgenic flies. (**A**,**B**) Transgenic flies treated with different concentrations of migalastat were analyzed for climbing ability at days 15 (**A**) or 22 (**B**) post-eclosion. The results are the mean ± SEM of three independent experiments. The expression of the α-Gal A mutant variants was under the dopa decarboxylase GAL4 driver (DdcGAL4). Statistical analysis included three-way ANOVA (to show the significance of the reaction of the different genotypes to different migalastat concentrations and test if there is a difference between the different days evaluated), followed by one-way ANOVA with post-hoc Dunnett test. Significance: * *p* < 0.05; ** *p*< 0.01.

**Figure 5 ijms-21-07397-f005:**
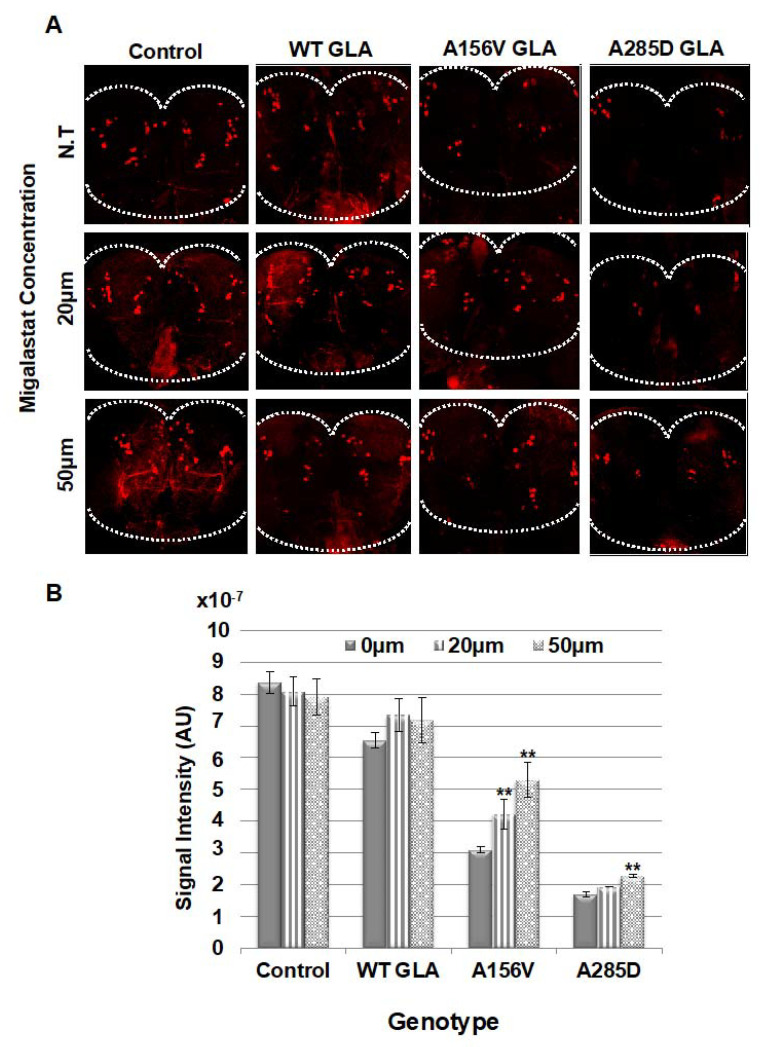
Immunostaining of dopaminergic cells in *GLA* transgenic flies. (**A**) Representative confocal images of adult brains of transgenic flies, expressing the human WT or mutant *GLA* variants, untreated or migalastat treated, stained with anti-TH antibodies at day 22 post-eclosion. (**B**) Quantification of TH signal intensity obtained from six tested brains as shown in (**A**). The expression of the α-Gal A mutant variants was under the dopa decarboxylase GAL4 driver (DdcGAL4). Statistical analysis included two-way ANOVA (to determine significant differences between genotypes’ reaction to concentrations) followed by one-way ANOVA with post-hoc Dunnett t-tests. Significance: ** *p* < 0.01. N.T: Non-Treated.

**Figure 6 ijms-21-07397-f006:**
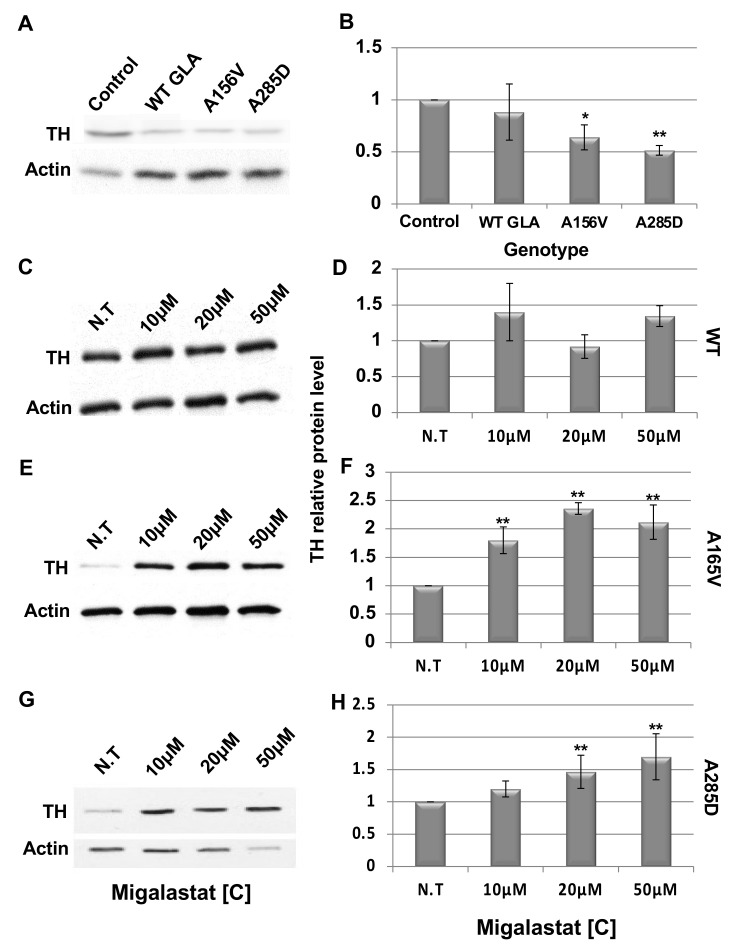
Western blot analysis of tyrosine hydroxylase in brains of *GLA* transgenic flies. (**A**) Protein lysates, prepared from heads of 10 control flies or flies expressing the different α-Gal A variants at day 22 post-eclosion, were subjected to Western blotting. The corresponding blots were interacted with anti-TH antibodies. As a loading control, the blots were interacted with anti-actin antibodies. (**B**) To quantify the results, TH intensity in each lane was divided by that of actin in the same lane, and the number obtained for Control was considered 1. Results represent the mean ± SEM of four independent experiments. Statistical analysis included one-way ANOVA with post-hoc Dunnett test. (**C**,**D**) Protein lysates prepared from flies expressing the WT α-Gal A, treated with the shown migalastat concentrations, were manipulated as in (**A**). To quantify the results, TH intensity in each lane was divided by that of actin in the same lane, and the number obtained for N.T. was considered to be 1. Results represent the mean ± SEM of four independent experiments. (**E**,**F**) Protein lysates prepared from flies expressing the A156V α-Gal A, treated with the shown migalastat concentrations, were manipulated as in (**A**). Quantification was as in (**D)**. (**G,H**) Protein lysates prepared from flies expressing the A285D α-Gal A, treated with the shown migalastat concentrations, were manipulated as in (**A**), and quantification of results was as in (**D**). In (**D**,**F**,**H),** statistical analysis included two-way ANOVA (to determine significant differences between genotypes’ reaction to concentrations) followed by one-way ANOVA with post-hoc Dunnett t-test. Significance: * *p* < 0.05; ** *p* < 0.01. N.T: Non-Treated. The expression of the α-Gal A mutant variants was under the dopa decarboxylase GAL4 driver (DdcGAL4).

**Figure 7 ijms-21-07397-f007:**
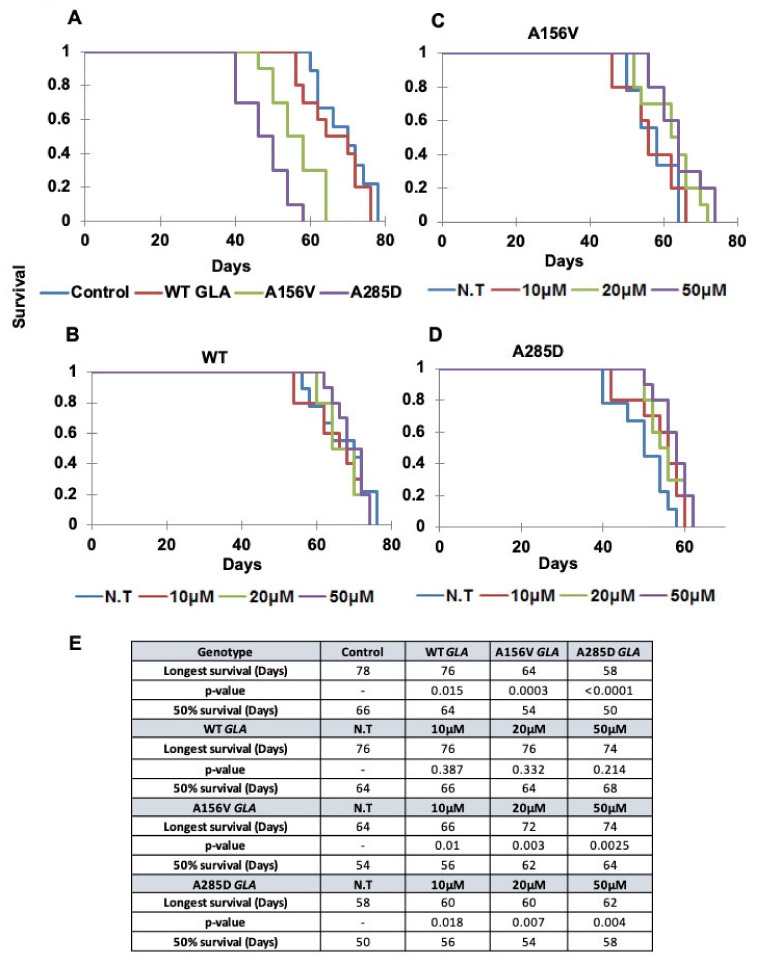
Survival of flies. (**A**) Kaplan–Meier curve showing the overall survival rates of flies expressing different *GLA* transgenes compared to normal control (Oregon-R) flies. (**B**–**D**) Kaplan–Meier curve showing the overall survival rates of flies expressing the different α-Gal A variants, treated with different migalastat concentrations. The expression of the α-Gal A mutant variants was under the dopa decarboxylase GAL4 driver (DdcGAL4). N.T.: non-treated. (**E**) A table showing the day reached by 50% of the flies (50% survival) and the longest survival time (days) reached by the different lines as well as statistical significance measured by Kaplan–Meier’s multiple comparisons. In this test statistical significance is *p* < 0.01 as compared to the wild type (**A**) or N.T. (**B**–**D**).

**Table 1 ijms-21-07397-t001:** Primers used for Gibson assembly.

Primer Name	Primer Sequence
FP-*GLA*(ATG) + pUAST/mycHis-left arm	5′-CAGATCTGCGGCCGCGGCTCGAGGATGCAGCTGAGGAACCCAGAACTAC-3′
RP-*GLA*(TAA) + pUAST/mycHis-right arm	5′-GCCCTCTAGAGGTACCCTCGAGCCAAGTAAGTCTTTTAATGACATCTG-3′
FP-*GLA*(ATG) + pcDNA4/myc-His-B-left arm	5′-CAGCACAGTGGCGGCCGGTCGAGTATGCAGCTGAGGAACCCAGAA-3
RP-*GLA*(TAA) + pcDNA4/myc-His-B-right arm	5′-CCGCGGGCCCTCTAGACTCGAGCGAAGTAAGTCTTTTAATGACATC-3′

**Table 2 ijms-21-07397-t002:** Primers used for qRT-PCR.

Primer Name	Primer Sequence
ATF4 F	5′-AGACGCTGCTTCGCTTCCTTC-3′
ATF4 R	5′-GCCCGTAAGTGCGAGTACGCT-3′
Hsc 70-3 F	5′-GCTGGTGTTATTGCCGGTCTGC-3′
Hsc 70-3 R	5′-GATGCCTCGGGATGGTTCCTTGC-3′
Xbp1 F	5′-CCGAACTGAAGCAGCAACAGC-3′
Xbp1 R	5′-GTATACCCTGCGGCAGATCC-3′
RP49 F	5′-TAAGAAGCGCACAAAGCACT-3′
RP49 R	5′-GGGCATCAGATATTGTCCCT-3′

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
