# Peer review of "Misfolding of Lysosomal α-Galactosidase a in a Fly Model and Its Alleviation by the Pharmacological Chaperone Migalastat"

_ijms, 2020, doi:10.3390/ijms21197397_

Round 1
Reviewer 1 Report
The authors employed a Drosophila system to test ER Associated degradation (ERAD) for Fabry disease. The merit of this model is to demonstrate a therapeutic effect of a chaperone on this creature. There are a few technical problems for this manuscript:
- The authors have tested a few alpha-Gal mutants. Were these mutant tested by the Migalastat in vitro assay?
- The authors asked whether mutant α-Gal A variants undergo ERAD with the proteasome inhibitors MG132. I don’t think the MG132 assay answers the question.
- alpha-Gal level drops at 50 uM Migalastat for A285D, this makes us worry that a high-dose Migalastat can be harmful.
- Figure 4 seems to be lost
- Figure 6, the tyrosine hydroxylase data are poor. A285D seems to be more preserved in TH and with a better response to Migalastat, than in other assays. Can you explain?
- The survival data on Figure 7 don’t look convincing. Can you show p values?
Author Response
ANSWERS TO REVIEWERS
We appreciate and thank the reviewers for their valuable comments, according to which we corrected the manuscript.
REVIEWER 1
Comments and Suggestions for Authors
The authors employed a Drosophila system to test ER Associated degradation (ERAD) for Fabry disease. The merit of this model is to demonstrate a therapeutic effect of a chaperone on this creature.
There are a few technical problems for this manuscript:
- The authors have tested a few alpha-Gal mutants. Were these mutant tested by the Migalastat in vitro
These mutations were tested in-vitro by one of the authors of the present manuscript (Lukas, J., et al.: “Functional characterisation of alpha-galactosidase A mutations as a basis for a new classification system in Fabry disease”. PLoS Genet 2013, 9, e1003632). The A156V variant had 4.3% in-vitro residual activity (of WT), while the A285D mutant had no detectable activity, in HEK293 cells transfected with plasmids expressing the different mutations. Furthermore, A156V variant was classified as migalastat responsive while A285D was not responsive. The latter had an insignificant α-Gal A activity change after migalastat treatment.
- The authors asked whether mutant α-Gal A variants undergo ERAD with the proteasome inhibitors MG132. I don’t think the MG132 assay answers the question.
MG132 assay is widely used to test ERAD of mutant variants (for example: Kelly et al: “Regulation of Ubiquitin-Proteasome System–mediated Degradation by Cytosolic Stress” Mol. Biol. Cell 2007, 18, 4279-4291; Tiengwe et al: “Variant Surface Glycoprotein, Transferrin Receptor, and ERAD in Trypanosoma brucei”. Cell Microbiol 2016, 18, 1673-1688).
We presented the MG132 assay in addition to an endo-H sensitivity assay (Figure 1C,D), which strongly indicated ERAD of the mutant variants used in this study.
- alpha-Gal level drops at 50 mM Migalastat for A285D, this makes us worry that a high-dose Migalastat can be harmful.
The reviewer is right, 50mM of migalastat may be harmful. Interestingly, it did not affect survival of the WT flies. Our aim was to test the effect of migalastat on decrease of UPR parameters so we used concentrations employed by others, for example: Lukas et at, 2020 (Seemann et al.: “Proteostasis regulators modulate proteasomal activity and gene expression to attenuate multiple phenotypes in Fabry disease”. Biochem J 2020, 477, 359-380).
- Figure 4 seems to be lost.
We apologize for loading twice Figure 3 and not presenting Figure 4.
The correct Figure 4 has been uploaded in the revised manuscript.
- Figure 6, the tyrosine hydroxylase data are poor. A285D seems to be more preserved in TH and with a better response to Migalastat, than in other assays. Can you explain?
We think that the blots showing tyrosine hydroxylase data are clean and clear. The reviewer is right, there was a maximal 60% change in the amount of TH in brains of A285D flies treated with 50 mM migalastat while there was a 30% increase in the amount of TH positive cells in these brains as presented in Figure 5 and no significant change in UPR markers after migalastat treatment. It is of note that different assays (i.e.: quantitative real time PCR, western blotting) do not produce exactly the same results. All in all, the results strongly indicated that the A285D mutant a GAL A variant is not amenable to migalastat treatment.
Reviewer 2 Report
Misfolding in Fabry disease is an interesting topic. As mentioned in the introduction, authors have replicated their successful protocol from Gaucher disease to Fabry disease and test whether the chaperone therapy could have an effect on the suspected disorder.
Introduction
Line 52-54: Authors should complete their description of Fabry disease and clarify amenability for migalstat: In males, two different phenotypes: classic Fabry disease with enzymatic activity < 3%, frequently associated to non sense mutation, frequently non amenable for Migalastat; Non classic Fabry disease or cardiac Fabry disease with enzymatic activity 3-30% due to missense mutation possibly accessible to Migalastat (amenable mutations). This point appears essential because authors use A156V mutation (amenable) and A285D mutation (non amenable) with possible different “misfolding characteristics” and use DGJ as a surrogate for migalastat.
Fabry women can also be very severely affected.
Line 69: migalastat increase enzyme activity when mutation is amenable only
Methods:
Authors use ANOVA parametric test that should not be use for such small samples with probable non Gaussian variables. Authors should use Kruskal Wallis’ test.
Results
- Figure 1E: please clarify why authors use ERK as loading control whereas Erk is modified by MG132. This WB is not very satisfying… Line 140 please change “results shown in E” and not “in D”.
- Line 150-151 and Figure 2C/D: authors can not affirm such conclusion about trafficking on this experiment! Authors should temper the message
- Line 179-180: authors affirm that “there was non significant change in UPR parameters (…) with A285D = This suggest that migalstat does not play any role on A285D variant in terms of misfolding
- Paragraph 2.3 and 2.4: this refers to experiments applicable to Gaucher disease and parkinsonism. The climbing activity of Fabry flies and their dopaminergic brain content appear very far from the problematic of Fabry disease. It appears difficult to interpret and extrapolate these findings to Fabry pathophysiology.
Figure 4 is the same as Figure 3.. I cannot see any fly climbing…
Line 220: authors mentioned that there is no effect of Migalastat on WT mutant nevertheless, the 50µM shows a significant increase. It has to be clarified.
Discussion:
Line 294: amenability: to be developed
Line 298: OK A156V flies are better with Migalastat. Nevertheless, once more authors jump very (too) fast to the conclusion that this is due to UPR correction. The improvement may be due to the increase of enzymatic activity only! The experiment should have been controlled with enzyme replacement therapy.
Line 392: Results cannot be applied to all mutations! The disorder cannot be “relieved by chaperone” in non amenable mutations as authors describe in figure3 with A285D!
Author Response
ANSWERS TO REVIEWERS
We appreciate and thank the reviewers for their valuable comments, according to which we corrected the manuscript.
REVIEWER NO 2
Comments and Suggestions for Authors
Misfolding in Fabry disease is an interesting topic. As mentioned in the introduction, authors have replicated their successful protocol from Gaucher disease to Fabry disease and test whether the chaperone therapy could have an effect on the suspected disorder.
Introduction
Line 52-54: Authors should complete their description of Fabry disease and clarify amenability for migalstat: In males, two different phenotypes: classic Fabry disease with enzymatic activity < 3%, frequently associated to non sense mutation, frequently non amenable for Migalastat; Non classic Fabry disease or cardiac Fabry disease with enzymatic activity 3-30% due to missense mutation possibly accessible to Migalastat (amenable mutations). This point appears essential because authors use A156V mutation (amenable) and A285D mutation (non amenable) with possible different “misfolding characteristics” and use DGJ as a surrogate for migalastat.
Fabry women can also be very severely affected.
We thank the reviewer for the clarification, and following the reviewer’s comment we added the details about Fabry disease and amenability of mutations to migalastat.
Since the name: “Migalastat” is widely used in the literature (McCaffery and Scott: “Migalastat: A Review in Fabry Disease”. Drugs (2019) 79:543–554; Germain et al: ”Treatment of Fabry’s Disease with the Pharmacologic Chaperone Migalastat”. N Engl J Med 375, 6, 545-555; Hughes et al: “Oral pharmacological chaperone migalastat compared with enzyme replacement therapy in Fabry disease: 18-month results from the randomized phase III ATTRACT study”. J Med Genet 2017, 54, 288–296), we decided not to change it.
Line 69: migalastat increase enzyme activity when mutation is amenable only.
According to the reviewer’s comment we added that migalastat increases enzyme activity of amenable mutations in cell culture and in mice.
Methods:
Authors use ANOVA parametric test that should not be use for such small samples with probable non Gaussian variables. Authors should use Kruskal Wallis’ test.
We added more details about the statistical analysis carried out in the present study. It is of note that the Kruskal-Wallis one-way ANOVA is a non-parametric method for comparing k independent samples. It is roughly equivalent to a parametric one-way ANOVA with the data replaced by their ranks.
For the fly experiments the samples are not small (100 flies/survival assay, 30 flies/climbing assay). As far as quantification of western blots one-way ANOVA is widely used in the literature (Seemann et al.: “Proteostasis regulators modulate proteasomal activity and gene expression to attenuate multiple phenotypes in Fabry disease”. Biochem J 2020, 477, 359-380; Ordonez et al: “Alpha-Synuclein induces Mitochondrial Dysfunction through Spectrin and the Actin Cytoskeleton”. Neuron 2018, 97, 108-124).
Results
- Figure 1E: please clarify why authors use ERK as loading control whereas Erk is modified by MG132. This WB is not very satisfying.
Total ERK does not change as a result of MG132 treatment, rather phospho-ERK changes (See: Choi et al: “Proteasome inhibition-induced p38 MAPK/ERK signaling regulates autophagy and apoptosis through the dual phosphorylation of glycogen synthase kinase 3b”. BBRC 2012, 418, 759-764).
Line 140 please change “results shown in E” and not “in D”.
The mistake was corrected
- Line 150-151 and Figure 2C/D: authors cannot affirm such conclusion about trafficking on this experiment! Authors should temper the message
We stated in the text that: “There was a 2.5 and a 3 fold increase in the total amount of the A156V α-Gal A following 10μM and 50μM, chaperone treatment, respectively, and elevation in the lysosomal fraction of the mutant protein, indicating that migalastat is able to bind the protein and to assist in its folding and trafficking from the ER to the lysosomes (Figure 2C, D)”.
Our results showed increase in the lysosomal fraction of A156V α-Gal A variant following treatment with migalastat. Such increase can occur only via trafficking from the ER to the lysosomes.
- Line 179-180: authors affirm that “there was non significant change in UPR parameters (…) with A285D = This suggest that migalstat does not play any role on A285D variant in terms of misfolding.
The reviewer is completely right, migalastat cannot correct the severe misfolding of A285D α-Gal A variant.
- Paragraph 2.3 and 2.4: this refers to experiments applicable to Gaucher disease and parkinsonism. The climbing activity of Fabry flies and their dopaminergic brain content appear very far from the problematic of Fabry disease. It appears difficult to interpret and extrapolate these findings to Fabry pathophysiology.
The reviewer is completely right, it is difficult to interpret and extrapolate the presented findings to Fabry disease pathophysiology. However, we did not use climbing ability of the flies or the amount of TH in their brain as any marker associated with Fabry disease. We used them as outputs of UPR, and tested changes in these parameters as the result of migalastat treatment.
Figure 4 is the same as Figure 3. I cannot see any fly climbing…
We thank the reviewer and apologize for the mistake of uploading twice Figure 3. We have now added the correct Figure 4.

Round 2
Reviewer 1 Report
no more comment
Author Response
REVIEWER 1
no more comment

Reviewer 2 Report
Response
It is of note that the Kruskal-Wallis one-way ANOVA is a non-parametric method for comparing k independent samples. = YES. A good non parametric test is better than an improperly used parametric one.
It is roughly equivalent to a parametric one-way ANOVA with the data replaced by their ranks. = NO !
To use an ANOVA you are supposed to check the that the experimental errors of your data are normally distributed. The variances between samples are equals with h omogeneity of variances and homoscedasticity. You are supposed to give information about the independence of samples.
As far as quantification of western blots one-way ANOVA is widely used in the literature= That is true and sad. This is a reason why medical science is less and less credible
Figure 1E: "Total ERK does not change as a result of MG132 treatment, rather phospho-ERK changes "If it does not change, how do you explain the obvious difference of ERK between MG132+ and - ? Authors have to clarify it.
Author Response
REVIEWER 2
Comments and Suggestions for Authors
- It is of note that the Kruskal-Wallis one-way ANOVA is a non-parametric method for comparing k independent samples. = YES. A good non parametric test is better than an improperly used parametric one.
It is roughly equivalent to a parametric one-way ANOVA with the data replaced by their ranks. = NO !
To use an ANOVA you are supposed to check the that the experimental errors of your data are normally distributed. The variances between samples are equals with homogeneity of variances and homoscedasticity. You are supposed to give information about the independence of samples.
As far as quantification of western blots one-way ANOVA is widely used in the literature. That is true and sad. This is a reason why medical science is less and less credible.sometimes.
Based on the reviewer’s comments, and not being real statisticians, we ended having an expert statistician (Dept of Statistics, School of mathematics at Tel Aviv University) analyze our results. The statistician went through the raw data and explained us that according to the central limit theory, normal distribution is assumed for all our experiments and therefore ANOVA tests are accepted. The statistician performed one-way, two-or three-way ANOVA tests with post-hoc Dunnett or one-sample t-tests. All the performed tests are detailed in the Methods section and in every figure legend.
- Figure 1E: "Total ERK does not change as a result of MG132 treatment, rather phospho-ERK changes "If it does not change, how do you explain the obvious difference of ERK between MG132+ and - ? Authors have to clarify it.
ERK serves as a normalizing control, since the amount of protein loaded in the different lanes is different, although we load samples containing what we think is the same amount of protein. In Fig 2E, the tested protein is α-Gal A before and after MG132 treatment. The ERK values are the same for: WT, +/- treatment, A156V +/- treatment and they are different for the A285D mutant, which means not the same amount of protein was loaded in the two lanes, +/- MG132. Though there is less ERK in the treated sample in comparison to the non-treated one, there is more A285D α-Gal A protein in the treated sample. After correction to the amount of ERK (as explained in the legend to the figure) it rises even more, as shown in the quantification in Fig 1F.